# Personalized Machine Learning-Based Prediction of Wellbeing and Empathy in Healthcare Professionals

**DOI:** 10.3390/s24082640

**Published:** 2024-04-20

**Authors:** Jason Nan, Matthew S. Herbert, Suzanna Purpura, Andrea N. Henneken, Dhakshin Ramanathan, Jyoti Mishra

**Affiliations:** 1Neural Engineering and Translation Labs, University of California San Diego, La Jolla, CA 92093, USA; srpurpura@health.ucsd.edu (S.P.); dramanathan@health.ucsd.edu (D.R.); jymishra@health.ucsd.edu (J.M.); 2Department of Bioengineering, University of California San Diego, La Jolla, CA 92093, USA; 3Department of Psychiatry, University of California San Diego, La Jolla, CA 92093, USA; m1herbert@health.ucsd.edu; 4Department of Mental Health, VA San Diego Medical Center, San Diego, CA 92161, USA; andrea.henneken7@gmail.com; 5Center of Excellence for Stress and Mental Health, VA San Diego Medical Center, San Diego, CA 92161, USA

**Keywords:** machine learning, healthcare professionals, empathy, wellbeing, N-of-1 model, EMA

## Abstract

Healthcare professionals are known to suffer from workplace stress and burnout, which can negatively affect their empathy for patients and quality of care. While existing research has identified factors associated with wellbeing and empathy in healthcare professionals, these efforts are typically focused on the group level, ignoring potentially important individual differences and implications for individualized intervention approaches. In the current study, we implemented N-of-1 personalized machine learning (PML) to predict wellbeing and empathy in healthcare professionals at the individual level, leveraging ecological momentary assessments (EMAs) and smartwatch wearable data. A total of 47 mood and lifestyle feature variables (relating to sleep, diet, exercise, and social connections) were collected daily for up to three months followed by applying eight supervised machine learning (ML) models in a PML pipeline to predict wellbeing and empathy separately. Predictive insight into the model architecture was obtained using Shapley statistics for each of the best-fit personalized models, ranking the importance of each feature for each participant. The best-fit model and top features varied across participants, with anxious mood (13/19) and depressed mood (10/19) being the top predictors in most models. Social connection was a top predictor for wellbeing in 9/12 participants but not for empathy models (1/7). Additionally, empathy and wellbeing were the top predictors of each other in 64% of cases. These findings highlight shared and individual features of wellbeing and empathy in healthcare professionals and suggest that a one-size-fits-all approach to addressing modifiable factors to improve wellbeing and empathy will likely be suboptimal. In the future, such personalized models may serve as actionable insights for healthcare professionals that lead to increased wellness and quality of patient care.

## 1. Introduction

Due to the demanding nature of their work, healthcare professionals often carry a significant psychological load. Burnout, typically characterized as emotional fatigue, depersonalization, and a diminished sense of personal achievement, is prevalent among healthcare professionals, particularly physicians. One in three physicians report burnout [1] and physicians express higher levels of dissatisfaction with work–life balance compared to other workers in the U.S. [2], which results in decreased patient satisfaction and an increase in medical errors [3,4]. Further, burnout is consistently associated with decreased empathy and has a direct negative correlation with the amount of compassion felt for others [5,6]. The wellness of healthcare workers has been further strained by the COVID-19 pandemic, with consistent reports of heightened stress, anxiety, and depression compared to pre-pandemic levels [7]. As such, it is important to mitigate burnout, improve wellness in healthcare workers, and identify ways to increase empathy towards patients, leading to better patient care [8,9,10,11].

From the perspective of the healthcare worker, there are uncontrollable and modifiable factors that contribute to burnout and decreased wellness. Uncontrollable factors include socioeconomic status, long work hours, delayed gratification, and the difficulty of their jobs, which all subsequently affect their work–life balance [12,13]. Modifiable factors related to burnout and wellbeing include social support, diet, sleep, and exercise, along with mental health conditions like depression and anxiety [14,15]. Given the strong relations between lifestyle attributes, burnout, and empathy, it could prove useful to build models exploring these interactions and better understand underlying mechanisms.

Various intervention approaches have been examined in an attempt to improve wellbeing and empathy in medical professionals; a 2021 review focused on investigating burnout and respective treatment plans in palliative care found 10 studies with various intervention plans including programs that span spiritual practices/meditation, mindfulness, communications skills, art therapy, educational, and physical activity [16]. Of these 10 studies, only 6 of them were successful in reducing various aspects of burnout post-study. However, there was no follow-up to measure if the treatment plans had a sustained effect. A similar review on improving empathy through training found 19 studies that were able to significantly improve levels of empathy in healthcare workers [17]. The training used focused on education, communication skills, perspective-taking, psychotherapy, direct empathy skills, arts and humanities, mindfulness, and gaming intervention. Over half of the studies also provided follow-up surveys at 12 weeks and found a smaller significant effect size. Some studies have provided more vigorous interventions in an attempt to combat burnout. For example, in a 2021 study, 76 physicians participated in an 8-week active mindfulness training program tailored to physicians followed by a 4-month maintenance phase that consisted of three monthly booster sessions [18]. A significant decrease in burnout at the end of the study period was found compared to a control group at the end of the maintenance phase; however, at the 12-month follow-up, there was no significant decrease in observed burnout.

Notably, these previous studies have focused on assessing and intervening at the group level, which is important for examining population trends and the average impact of the intervention. However, it is important to recognize that what triggers and maintains burnout likely varies at the individual level. For example, whereas inadequate social support may be most related to poor wellbeing in one individual, for another individual with strong social support, poor wellbeing may be most related to anxious mood or poor sleep. Individuals may also be less receptive to an intervention if that program does not apply to their unique life circumstances. For these reasons, personalized plans can inform the most effective avenues for intervention in each individual. Further, group-level interventions may also fall short because they are more challenging to sustain following the intervention, as seen in one study where burnout levels returned during the follow-up period [18]. Finding intervention plans based on a person’s attributes, however, can be much easier to sustain and yield higher satisfaction as seen in studies that have used personalized technologies in diverse cases of diabetes and hypertension management, amblyopic eye vision, and text communication for hospital patients [19,20,21,22].

In the current study, we aim to identify personalized models of wellbeing and empathy in healthcare professionals using mood and lifestyle data attributes collected via ecological momentary assessments (EMAs) and passive data collected from wearables over a 30-day study period. Theoretically, wellbeing goes beyond the absence of distress/burnout and includes feelings of thriving [23,24]. Especially for healthcare professionals, high personal wellbeing makes them more attentive to their patients’ experience and, thereby, may enhance empathy towards their patients [25]. For prediction, we focus on mood and lifestyle variables that have been previously associated with burnout, wellbeing, and/or empathy, including depression, anxiety, mindfulness, diet, sleep, exercise, social relationships, and gratitude [14,18,26,27,28,29,30,31]. This idiographic personalized modeling approach has been previously validated, examining individualized models of depressed mood [32]. This approach is user-friendly, readily available, and enables the gathering of real-world data, potentially offering more comprehensive and precise insight into factors most closely associated with individual wellbeing and empathy. Personalized modeling may play an important role in pinpointing elements that enhance or diminish wellbeing and empathy, which can in turn guide preventive measures as well as individualized intervention strategies, with the ultimate goal of boosting wellness in healthcare professionals. We hypothesize that this data-driven approach will be able to firstly provide accurate predictive models for wellbeing and empathy in physicians and show diversity in top predictors across models highlighting the need for personalized treatment plans.

## 2. Materials and Methods

### 2.1. Participants

A total of 12 healthcare professionals participated in the study (mean age: 28 ± 3.1, range: 24–35 years, 4 males). Participants were recruited from the UCSD School of Medicine using email and campus flyers. All participants were fluent in English and healthy adults, i.e., did not have any current medical diagnosis nor were taking any current psychotropic medications. All participants gave written informed consent in accordance with the Declaration of Helsinki before participating in the study. All experimental procedures were approved by the Institutional Review Board of the University of California San Diego (UCSD) (protocol #180140). Data collection took place during Spring 2021–Fall 2022.

### 2.2. Study Procedure

On day 1, participants downloaded the Unity-based *BrainE* application on their iOS/Android smartphone [33]. Within the *BrainE* app, participants accessed daily EMAs on a module called MindLog, on which they provided mood and lifestyle ratings once per day for the duration of the study (a total of 60 sessions, up to 3 months). The app sent regular notifications daily to all participants following the methodology of recent research on longitudinal mood monitoring [32,34]. Participants also received a Samsung Galaxy wristwatch on day 1 that they wore throughout the study, except while charging the watch for a few hours once every 2–3 days.

### 2.3. Wellbeing and Empathy Ratings

Using EMA, participants rated their personal wellbeing and their empathy towards patients on a 14-point Likert scale shown as a red-to-green color gradient. For wellbeing, participants responded to “How is your present mental wellbeing?” with the “Thriving” label anchored to the score of 14 shown in green, and the “Burned-out” label anchored to the score of 1. The use of a Likert scale for the responses in the study was chosen because we did not want to burden the user with lengthy symptom surveys at each session for up to 60 sessions. Also, Likert scales have been standardly applied for longitudinal EMA-based mood/behavior monitoring in past research [34,35,36,37,38,39].

For empathy, participants responded to “How much tender concern do you feel for your patients today?” shown as a red-to-green color gradient with the “Full Concern” label anchored to the score of 14 shown in green and the “None” label anchored to the score of 1 shown in red.

Wellbeing and empathy were the main dependent variables that we were interested in predicting. Out of the total 12 participants, all completed the wellbeing EMA and 7 completed the empathy EMA, as the others were healthcare staff who did not have direct daily interactions with patients.

### 2.4. Mood Ratings

Participants also rated depression and anxiety on 14-point Likert scales shown as green-to-red color gradient scales. For depression, participants responded to “How happy vs. sad/ depressed do you feel right now?” with the “Happy” label anchor next to the score of 1 and the “Sad or Depressed” label anchor next to the score of 14. For anxiety, participants responded to “How relaxed vs. anxious do you feel right now?” with the “Relaxed” label anchor next to the score of 1 and the “Anxious” label anchor next to the score of 14. Gratitude was also rated, on a 1–7 Likert scale with the prompt being “Take a moment to indicate how grateful you are feeling.”; participants were given 7 icons graded from rainy weather to sunny weather to choose from.

### 2.5. Interoceptive Attention to Breathing Assessment

At each EMA, participants completed a rapid 30 s assessment, Breathe, in which they were requested to tap the mobile screen after each full breath (inhale plus exhale). Recent research shows that such objective monitoring can serve as a basic assay of breath-focused attention related to mindfulness and inversely related to the internally distracted/ruminative state of the individual, which is exacerbated in depression [40,41,42]. Mean breathing time and consistency data were extracted on this assessment at each EMA.

### 2.6. Diet Reporting

At each EMA, participants reported their recent consumption of sugars, fats, and caffeine over the last 24 h. To improve compliance, we opted for a simplified version of diet reporting instead of more objective methodologies which can be burdensome [43,44]. Specifically, within the context of depression, the excessive consumption of processed fats and sugars has been related to the severity of symptoms, and intervention to change such diet patterns has shown success [45,46,47,48]. Hence, based on a standard assessment of dietary fats and sugars, we asked the following questions once per day, completed on a 0–12-item scale [49].

Fats: How many of these items have you had in the last 24 h? Red meat burger/sandwich; sausage/salami/bacon; whole egg; white bread; pizza; cheese; French fries; chips; butter popcorn; whole milk/milkshake; and fast-food take-out.

Sugars: How many of these items have you had in the last 24 h? Cake/cookies; ice cream; chocolate; candy; pancakes/French toast; jam/honey; soda; juice or other sweetened beverage; and cereal with added sugar.

Caffeine: How many servings of caffeine (coffee/tea/energy drink) have you had in the last 24 h?

Participants also rated their satisfaction with their overall diet on a 1–5-star system.

### 2.7. Sleep Reporting

For each daily EMA, participants reported their sleep habits of the night before, including sleep time, wake-up time, sleep duration, a percentage estimate of the time in bed spent asleep, and a 1–5-star rating on sleep satisfaction.

### 2.8. Exercise Reporting

For each daily EMA, participants reported the amount of exercise (in hours and minutes) they engaged in in the past 24 h in the following three categories:Strenuous exercise (e.g., running, vigorous sports, or bicycling)Moderate exercise (e.g., fast walking, easy bicycling, swimming, or dancing)Mild exercise (e.g., yoga or easy walking)

Participants also rated their satisfaction with their exercise on a 1–5-star system.

### 2.9. Social Connection Reporting

At each EMA, participants reported their overall social connection in the past 24 h. Questions include the following:How many people close to you did you talk to? 1–10+How much total time did you spend chatting? In hours and minutes.How long were you engaged in an organized group activity in-person or online (support/sports/exercise/hobby/professional group)? In hours and minutes.How long were you engaged in volunteer work for any organization in-person or online (religious, charitable, political, health-related)? In hours and minutes.

Participants also rated their satisfaction with their social connection on a 1–5-star system.

### 2.10. Active Reflection

Participants also participated in a brief active reflection task where they were prompted with one of 10 unique questions based on the literature on positive psychology and gratitude [50]. The question prompts were refreshed every 6 days. (1) Who or what made you smile? (2) Who or what are you thankful for? (3) Note a moment you enjoyed. (4) Note an act of kindness you did or observed. (5) Note a moment you found inspiring. (6) Who or what keeps you going? (7) Note a moment worth celebrating! (8) Everyone has personal strengths. Recognize one of yours. (9) Think of a challenge you faced, small or big, and what you learned from it. (10) Dedicate a note of appreciation to yourself or your loved one(s).

The total time spent on the module and the amount of time they spent typing out a response were recorded.

### 2.11. Screentime Reporting

At each EMA, participants reported their current and past day screentime as well as how much of that time was spent for social purposes (social media, messaging friends, etc.) in hours and minutes. They also rated their screentime satisfaction on a 1–5-star system.

### 2.12. Smartwatch Data

From the Samsung Galaxy smartwatch, we extracted features corresponding to (1) heart rate; (2) step count and exercise including speed, calories burned, distance, and duration. For all features, start and end times were extracted.

### 2.13. Machine Learning (ML) Models Training and Evaluation Strategy

The general pipeline architecture has been previously validated in our published research and modified for this study [32]. Key components are standard across ML analysis and include data ingesting and feature extraction, data preprocessing, and ML model training and evaluation. Notably, the present study adds an additional ML model and new feature sets that were not explored in the previous study, and it targets different outcomes.

### 2.14. Data Ingestion and Feature Extraction

The data from all the sources were carefully aligned, keeping in mind the different sampling rates of variables (seconds for smartwatch data up to days for EMA data). All independent data variables were either aggregated or extrapolated based on their sampling frequencies to match the sampling frequency of the dependent variable (DV), i.e., wellbeing and empathy EMA ratings as the reference standards. The following features were, thereby, extracted from the EMA and smartwatch data:


EMA Data


Time of the day when a particular DV was taken: (6:00, 10:00), (10:00, 14:00), (14:00, 18:00), (18:00, 23:59).Anxiety, depression, and gratitude ratings were completed at each time point when a DV rating was obtained.Attention to mean breathing time and consistency at each EMA when a DV rating was obtained.Total amount of fats, sugars, caffeine, and diet satisfaction in the last 24 h of each DV rating.Sleep time, wake-up time, sleep duration, percent estimate of time in bed spent asleep, and 1–5-star rating on sleep satisfaction in the past 24 h period prior to the DV rating.Exercise duration for each intensity type and total satisfaction in the 24 h period prior to the DV rating.Number of people and total time spent chatting, total time in an organized group, total time spent volunteering, and total satisfaction in the 24 h period prior to DV rating.Amount of time the response section of the active reflection module was open and actively used at each EMA session.


Smartwatch data


9.Heart rate was measured as the mean value from a window of ±30 min around the time of each DV rating.10.Cumulative step features were taken as the mean values from the past 12 h of each DV rating for each step feature separately.11.Cumulative exercise features were taken as the mean values from the past 24 h of each DV rating calculated for each feature separately.

The features that participants had no responses for throughout the entire study were considered missing and dropped for that participant. These features were calculated and stored separately for each subject for a max of 47 features possible per participant. The data were also inspected using both automated and manual approaches for unusable and missing variables, as well as variables with zero variance that were dropped for that participant. We did not implement any additional feature selection such as PCA which would dissociate variables from their physical attributes to preserve model interpretability. Variables that are based on logging time spent on the *BrainE* app (attention to breathing and active reflection) are more prone to outliers by leaving the app running or distracted use artificially inflating the time spent on each module. For these, outlier removal was performed by setting 3 median absolute deviation (MAD) criteria.

The manual inspection of the raw data was only used to validate the metadata such as file names, variable names, and data format differences that occur from different mobile operating systems and smartwatch versions.

### 2.15. Data Preprocessing for ML Models

All data processing and ML modeling was conducted in python3 using various libraries including numpy, pandas, sklearn, seaborn, matplotlib, scipy, smogn, and timeshap. After the preliminary cleaning mentioned above, the missing data were filled with a regression-based iterative imputation from the sklearn library. For personalized models, removing missing data can create unaccountable bias and lead to low accuracy on test data [51]. This imputation did not change the overall distribution of the dataset, and on average, only 5% of the missing data was imputed across all subjects with a min and max of 0.6% and 7.5%. The data were then scaled using standard scalers also using sklearn.

The preprocessing steps were also wrapped in a “pipeline object”. The key advantage this provides is ensuring the preprocessing statistics are only derived from the training set and transforming the testing set with these statistics. This avoids any potential data leakage into the testing set during the cross-validation stages and improves computational efficiency.

### 2.16. ML Pipeline

To ensure accurate models, we employ standard best practices for ML analysis including nested cross-validation, hyperparameter tuning, and model selection and evaluation. This was performed for each model independently. Since subjects have different compliance rates, the number of datapoints will not be consistent for each model. On average, 55 ± 7 of 60 total MindLog EMAs were completed per participant with a range of 42–60 EMA sessions across participants.

### 2.17. Data Augmentation and Edge Cases

Initial models were built for each participant using the data available. For the participants with less than 45 datapoints and poor performance, a synthetic minority over-sampling technique for regression with Gaussian noise (SMOGN) was applied, and a new model was created [52]. If a model only predicts a constant value and has one dominant feature, that feature is removed, and the model is re-run. This was performed so the variability in the output variable could be better represented by all available features instead of one dominant feature. Two participants (P-7 empathy and P-28 wellbeing) had constant predictions on the first run; however, only P-7 had one dominant Shapley feature, so it was re-run.

### 2.18. Cross Validation

With a limited amount of data per subject, and wanting to limit model overfitting, we opted for a nested CV approach instead of a single CV scheme with the only downside being increased computation cost and time [53]. Here, we specifically used a repeated fourfold CV scheme with ten repeats as the inner CV strategy and a simple fourfold CV scheme as the outer CV strategy for the overall nested CV scheme. The predicted data were generated from the single left-out fold from the fourfold CV to ensure no data leakage. More details on the nested CV algorithm can be found in prior validations of this ML pipeline [32].

We modeled individual wellbeing and empathy mood ratings separately using the various modalities of data, i.e., MindLog EMA data and smartwatch lifestyle data employing supervised ML regression models with hyperparameter tuning and trained over the nested CV scheme. Figure 1 shows the main steps of the pipeline; the pipeline compared multiple ML strategies for each subject including random forest, gradient boost, adaptive (Ada) boost, elastic net, support vector, Poisson regressor, and a Long Short-Term Memory (LSTM) model. The voting regressor that employs the best model from all the other strategies besides LSTM was also used. LSTM was not included since this model architecture differed from others relying on using past samples while the other regressors operated on individual discrete time points. Details on each ML can be found in our prior publication [32]. After hyperparameter tuning and training over all these ML models, the results were evaluated for each model, and each subject over the regression metrics of mean absolute percentage error (MAPE) and mean absolute error (MAE). We used MAPE as the performance metric to choose the best model (with the lowest error) for each ML strategy [32,54]. MAPE is calculated using the following formula:(1)$$MAPE=\frac{1}{n}\sum_{k=1}^{n}\left|\frac{P_{k}-A_{k}}{A_{k}}\right|\times 100$$
where *P_k_* is the predicted value of the kth datapoint, *A_k_* is the actual value of the *k*th datapoint, and *n* is the total number of datapoints.

We compared the outcome of the best-performing models from each strategy and calculated the overall best model with the least overall MAPE; we chose this model to represent each participant (see Appendix A for wellbeing models and Appendix A for empathy models). Thus, each study participant has up to two personalized models predicting their wellbeing and empathy.

### 2.19. Personalized ML Feature Importance

We also apply SHapley Additive exPlanations (SHAP) to each of the best fit models for model interpretability [32,55,56] These SHAP values indicate the relative importance and directionality of each feature as it predicts the outcome variable, allowing us to better understand the model architecture, i.e., how the model is making the outcome predictions. The result will reveal the specific lifestyle attributes that are most impactful for each individual.

## 3. Results

A total of 12 ML models were created separately for each participant, predicting their wellbeing scores, and 7 ML models were also created, separately targeting empathy scores for participants that have direct patient contact. There was a total of 47 possible variables for each subject across the EMA and smartwatch data. After removing missing variables and variables with zero variance, participants had an average of 33 ± 3.3 variables (range: 27–37). Appendix A show the number of independent variables, number of sessions, and the mean ± std MAPE of each type of model in the pipeline; the best fit of all models is also indicated for each individual in the table. The average MAPE of the individual best-fit models across all participants for wellbeing and empathy were 24.3 ± 12% and 13.6 ± 4%, respectively. This corresponds to a Mean Absolute Error (MAE) of 1.5 ± 1.6 and 1.2 ± 1.15, respectively, on the 14-point Likert scale.

Figure 2 shows the best model for each subject by MAPE, the distribution, and the heatmap of predicted vs. actual scores across all participants for wellbeing and empathy. Appendix A show individual predictions and histograms for wellbeing and empathy.

Figure 3 shows Spearman’s correlation between predicted and actual values for each subject for wellbeing and empathy. The columns with only wellbeing indicate subjects who did not have direct patient interactions to report empathy, and hence, only their wellbeing model was built. P-28 is missing the wellbeing datapoint due to the model generating a constant prediction possibly due to limited variance in the original input data. The overall Spearman’s rho for all participants were r(706) = 0.81, *p* < 1 × 10^−50^ and r(426) = 0.66, *p* < 1 × 10^−50^ for wellbeing and empathy, respectively.

Figure 4 and Figure 5 show the time-course predictions of wellbeing and empathy models as well as an overlayed histogram of predicted and actual values from all the time points. We observed that most of the histograms showed high overlap and actual vs. predicted values showed high correlation across time (Spearman’s values summarized in Figure 3 above).

To gain more insight into the model architecture, we computed Shapley statistics for each feature using the best ML model for each subject. Figure 4 shows the wellbeing and empathy model Shapley plots for participants with both dependent variables. Appendix A shows the remaining Shapley plots for participants with only the wellbeing variable. Feature rank importance and individual Shapley values are both shown.

Out of the seven participants who had both wellbeing and empathy models (Figure 4), four of them had empathy as a top predictor (i.e., among the top five) for their wellbeing model, and five had wellbeing as a top predictor (i.e., among the top five) in their empathy model. When looking at other common top predictors, 8/12 participants had depressed mood as a top predictor in their wellbeing model while only 2/7 participants had depressed mood as a top predictor in their empathy model. Anxiety was a top predictor for 8/12 and 5/7 wellbeing and empathy models, respectively. Features related to social connection were found in 9/12 wellbeing models but only 1/7 empathy models. Sleep attributes were seen in 4/12 wellbeing and 3/7 empathy models, exercise attributes in 2/12 wellbeing and 3/7 empathy models, diet attributes in 6/12 wellbeing and 1/7 empathy models, attention to breath in 2/12 wellbeing and 2/7 empathy models, and gratitude in 3/12 wellbeing and 1/7 empathy models, respectively. A visual representation of the percentages can be seen in Figure 5.

## 4. Discussion

Understanding factors that predict wellness and empathy at the individual level in healthcare professionals has significant implications for optimizing patient care while reducing burnout and supporting the wellbeing of the provider. In this study, we present a systematic PML pipeline that provided accurate predictions (average MAPE~10–20% across all best-fit models) of mental wellbeing and empathy towards patients based on mood and personal lifestyle data collected from smartphones and wearables. Notably, we were able to pinpoint the exact lifestyle attributes, such as sleep, exercise, diet, or social connection, that were most impactful for each individual. Notably, these features can be uniquely intervened on for each participant informing individual-adaptable wellness strategies tailored to each person’s lifestyle.

When examining the empathy and wellbeing models from the 7 participants who had both of these outcome reports, empathy and wellbeing were revealed as top predictors for each other in 9 of 14 (7 × 2) models, indicating a link between the two constructs. Additionally, it was always the case that higher wellbeing biased the model to predict higher empathy ratings and vice versa for empathy predicting wellbeing. In fact, the relationship between wellbeing and empathy in physicians is well studied and more nuanced than just establishing a link between them. One model of burnout suggests that higher empathy may help prevent burnout [59]. Although wellbeing and empathy are important predictors of each other, we found little overlap otherwise in the lifestyle variables. This is especially interesting since it may suggest that although wellbeing and empathy influence each other, they each have separate driving mechanisms.

Depressed mood was a top predictor of wellbeing in 8 of 12 participants, which aligns with the current understanding of the link between depression and wellbeing/burnout [14,60]. However, depression was only a top predictor of empathy in two of seven participants. There is less consensus on the link between depression and empathy. For example, among medical students, studies have found that depression was negatively associated with empathy [61,62], whereas another study found that depression was positively associated with empathic concern [63]. Anxiety, on the other hand, occurs at the same frequency for both wellbeing (8 out of 12) and empathy (5 out of 7) models. It has also been well documented that anxiety has a direct link to both wellbeing and feelings of empathy [26,64]. Despite the good characterization of anxiety and empathy, it is worth noting that one subject (P16) seemed to show an inverted relationship between anxiety and empathy; as seen in their Shapley plots (Figure 4), higher levels of anxiety were associated with greater patient empathy, which may not have been considered a possibility when looking at group-level analysis, though studies in nurses do suggest that greater anxiety can relate to greater personal warmth [65] and that in general, there is a positive relationship between social anxiety and affective empathy [66,67].

When considering lifestyle factors (sleep, diet, exercise, social connection), there was large variability among participants, with some overall trends between the wellbeing and empathy models. One notable observation is that variables relating to social connection were highly represented in wellbeing models, showing up in 9 of 12 participants compared to the empathy model where only 1 of 7 participants had a social connection variable as a top predictor. The significant role of social life in predicting physician wellbeing suggests that personal social life does impact their overall health, possibly by providing a necessary outlet for stress and contributing to a balanced work–life dynamic. Interestingly, the lack of overlap between social life and empathy could reflect the professional training physicians receive, which often emphasizes the development of empathy as a clinical skill, independent of personal life [59]. Furthermore, the interplay between social life and workplace environment warrants further investigation. A supportive workplace environment could foster social interactions and provide opportunities for physicians to decompress, thereby enhancing overall wellbeing. Conversely, a high-pressure, unsupportive environment could lead to burnout and reduced wellbeing. These findings underscore the complex relationship between personal and professional lives in healthcare settings, emphasizing the need for a holistic approach in devising strategies for physician wellbeing and patient care.

Group analysis, while useful in many contexts, has its limitations when it comes to individual healthcare [18,68]. Although there are general trends in our findings, the specific predictive features drastically vary across participants, which has direct implications for intervention. For example, although social connection was a consistent predictor of wellbeing, an intervention targeting this domain would likely be suboptimal for 3 of the 12 participants based on the directionality of the relationship between the identified top features and wellbeing. For example, increasing social connection attributes would potentially result in P-16 seeing a reduction in wellbeing since their model shows reduced chat time related to improved wellbeing.

Personalized treatments, tailored to an individual’s lifestyle, offer a variety of unique advantages in healthcare. By considering lifestyle factors, these treatments can be seamlessly integrated into a person’s daily routine. Individuals may be generally more receptive to modifications that align with their existing lifestyle, as opposed to an external program that may feel disruptive or overwhelming [69]. This approach may not only ensure that the treatment plan is realistic and achievable but it also increases the likelihood of adherence. Furthermore, the ability to make incremental adjustments can lead to sustainable lifestyle changes over time. Thus, personalized treatments represent a promising strategy in healthcare, fostering a patient-centered approach that respects individual lifestyle choices and promotes long-term health and wellbeing.

Overall, this PML research may offer an alternative to traditional approaches in managing burnout and improving empathy by helping to inform individualized intervention plans. Another notable strength of our approach is the scalability of both data collection and treatment planning using mobile devices and wearables. Another significant advantage of a personalized modeling approach is that it eliminates the need for one model that is generalizable across participants and populations, which is likely an impossible task [70]. Adopting personalized models would bypass the main problem with ML in healthcare, which is the question of generalizability across subjects.

There are also limitations to the current study. First, we tested the PML pipeline in a relatively small sample of research participants. Second, the pipeline is also susceptible to the quality of the data each participant provides. We also do see poor model performance in participants such as a near-constant predictor for one subject, possibly stemming from a lack of variability in the data collected. Further testing in a larger, more diverse cohort is required to help validate our methodology. It is also worth noting that research has indicated that interventions directed at healthcare workers are less effective compared to systematic changes in their respective organizations [71,72,73]. However, there is still validity in operating at the individual level because it is something that can be easily implemented.

Despite the limitations, this study offers a systematic and standardized way of identifying top features of wellbeing and empathy at the individual level among healthcare professionals. We found a high degree of feature variability across participants. Future research is needed to determine if tailored wellness and empathy interventions based on individualized modeling offer advantages over traditional one-size-fits-all approaches.

## Figures and Tables

**Figure 1 sensors-24-02640-f001:**
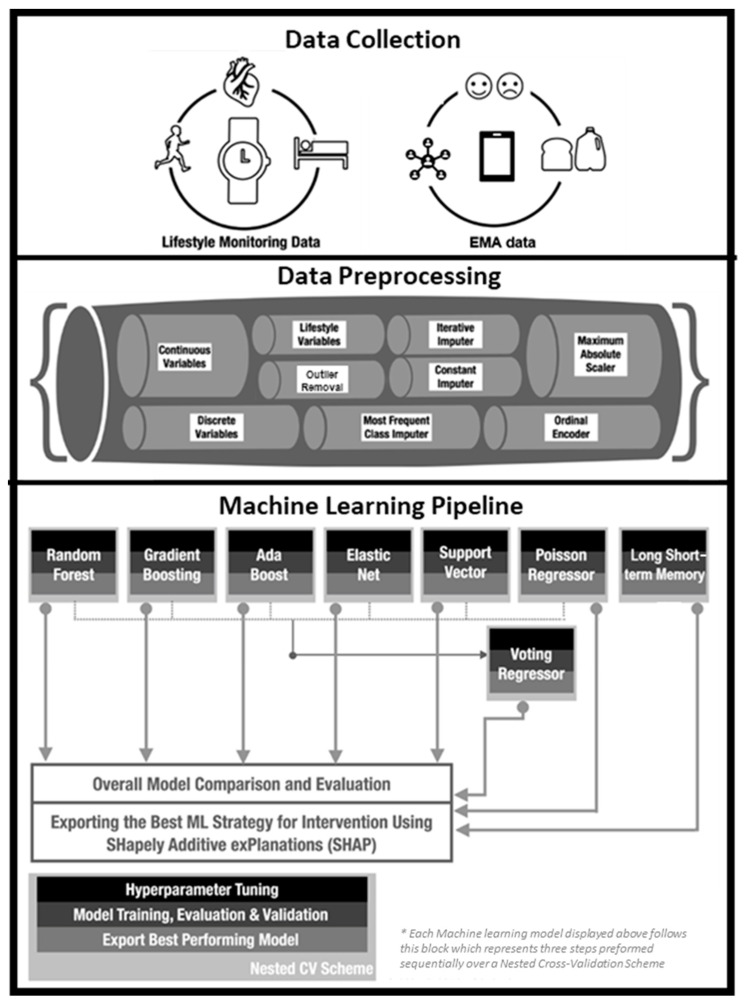
Personalized machine learning pipeline flowchart adapted from [32]. This pipeline is run for each subject with a target variable set to wellbeing and empathy.

**Figure 2 sensors-24-02640-f002:**
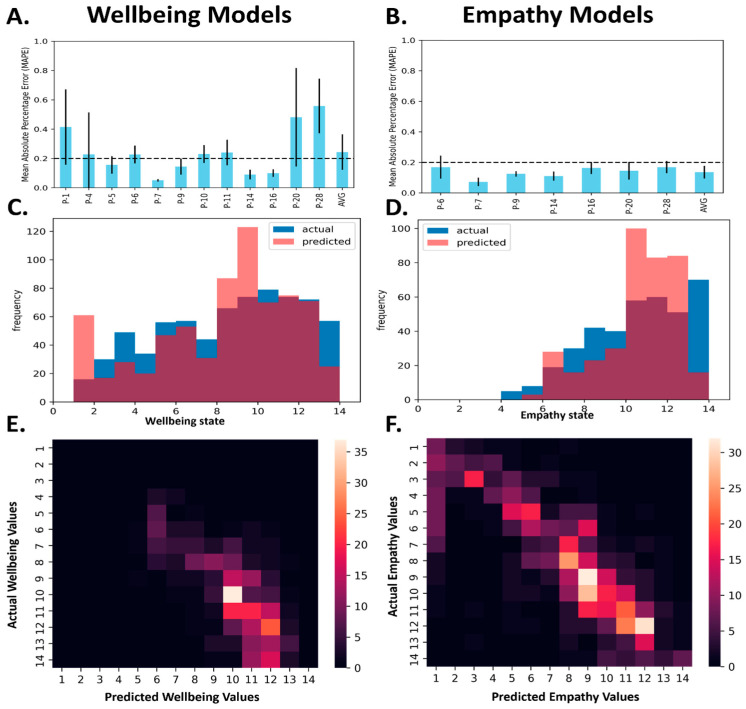
Results from wellbeing and empathy models. (**A**,**B**). Mean absolute percentage error (MAPE) of wellbeing and empathy models, respectively. Mean and std are shown from nested cross-validation for each subject and the average across all participants. The dashed line at 0.2 as a threshold for good MAPE [57]. (**C**,**D**). Combined histogram of all predicted values for wellbeing and empathy across all participants. Blue and red represents actual and predicted values respectively. (**E**,**F**). Heatmap of actual vs. predicted values for wellbeing and empathy.

**Figure 3 sensors-24-02640-f003:**
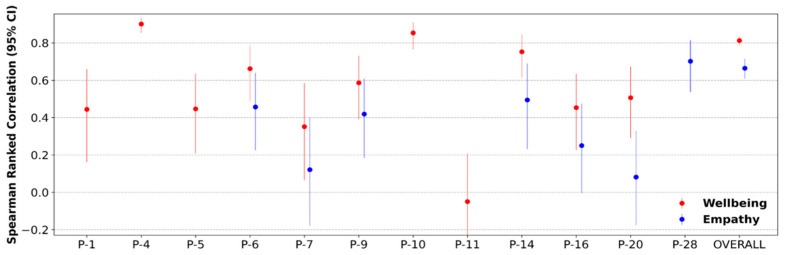
Spearman rank correlation coefficients with 95% confidence interval bounds for predicted vs. actual wellbeing and empathy scores. Composite Spearman’s rho was also calculated using all subject’s data indicated by the “overall” marker. In total, 10/11 and 4/7 participants showed significant correlations between predicted and actual data for wellbeing and empathy, respectively.

**Figure 4 sensors-24-02640-f004:**
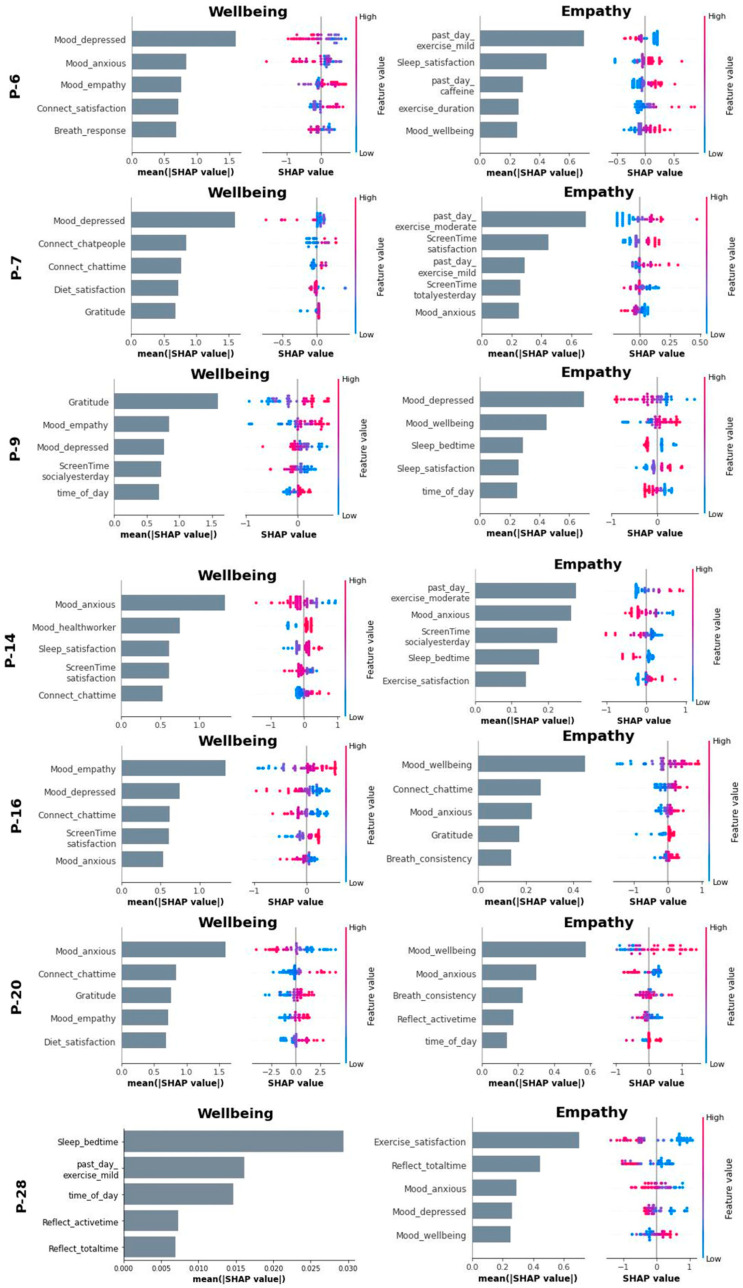
SHapley Additive Explainer (SHAP) plots across participants comparing wellbeing and empathy models. Rank feature importance and feature effects are shown. P28’s wellbeing slide shows timeshap [58] rank feature importance for their LSTM best-fit model. Individual effects of each feature on the dependent variable of wellbeing/empathy are depicted using the colored points to the right of the bar graph from low (blue) to high (pink) feature values relative to the outcome plotted on the *x*-axis (from low to high wellbeing/empathy going from left to right). For example, the P-6 Empathy Shapley plot shows that their top-ranking predictors of empathy are mild exercise over the past day followed by sleep satisfaction; since the color dots for mild exercise progress from pink to blue from left to right, lesser mild exercise supports greater patient empathy in this case but for sleep, greater sleep satisfaction (color dots progressing from blue to pink) supports greater patient empathy.

**Figure 5 sensors-24-02640-f005:**
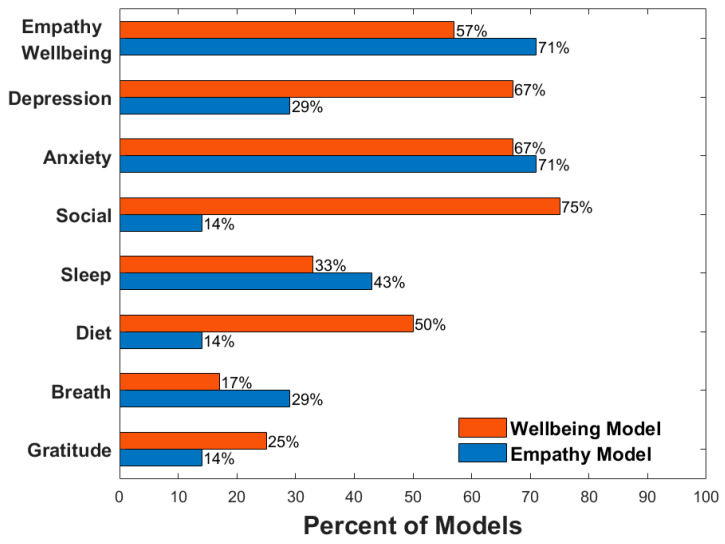
Percentages of models with specific mood and domain variables as top predictors, depression, anxiety, social connection, sleep, diet, breath attention, and gratitude. The first row, Empathy–Wellbeing, indicates percentages of wellbeing models with empathy as a top predictor and percentages of empathy models with wellbeing as a top predictor. Wellbeing model percentages are out of 12 total models except for the empathy variable which is out of 7. Empathy models are out of 7.

## Data Availability

De-identified and processed feature set data will be available upon request.

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
