# Peer review of "Personalized Machine Learning-Based Prediction of Wellbeing and Empathy in Healthcare Professionals"

_sensors, 2024, doi:10.3390/s24082640_

Round 1
Reviewer 1 Report
Comments and Suggestions for Authors
Please see attached file.

Reviewer 2 Report
Comments and Suggestions for Authors
The manuscript describes the personalised prediction of wellbeing and empathy in healthcare professionals and the overall manuscripts is well-organised. However, I have following comments.
- The abstract mentions the use of “SHAP for predictive insight into the model architecture”, however it is not clear in the manuscript how SHAP can provide insights into the model architecture.
- Introduction could have been improved by including theoretical basis for quantification of wellbeing and Empathy. Besides several covariates used in the study have widely used scales for measurement. They could be discussed too.
- In materials and methods, the basis of the selected Likert Scale is not clear. The reliability of the arbitrary scale based on direct self-response appears to be vague. Adoption of well-established measurement scales based on questionnaires would have improved the reliability of the data.
Round 2
Reviewer 1 Report
Comments and Suggestions for Authors
Thank you for addressing my comments. It's not clear if there was an issue with the conversion of the tracked changes Word document to PDF but there are numerous instances of mistakes in the edits. For example, line 233 that reads "The data from all the sources were carefully aggregatedaligned" and line 236 that reads "To reconcile these differences, allAll independent data". I encourage the authors and the editor(s) to review the final version for such mistakes prior to publication.
Comments on the Quality of English LanguageThank you for addressing my comments. It's not clear if there was an issue with the conversion of the tracked changes Word document to PDF but there are numerous instances of mistakes in the edits. For example, line 233 that reads "The data from all the sources were carefully aggregatedaligned" and line 236 that reads "To reconcile these differences, allAll independent data". I encourage the authors and the editor(s) to review the final version for such mistakes prior to publication.
Author Response
We thank the reviewer for bringing this to our attention. We have since integrated all the track changes to the manuscript and ensured there were no errors.